# Monitoring of Body Condition in Dairy Cows to Assess Disease Risk at the Individual and Herd Level

**DOI:** 10.3390/ani13193114

**Published:** 2023-10-06

**Authors:** Ramiro Rearte, Santiago Nicolas Lorenti, German Dominguez, Rodolfo Luzbel de la Sota, Isabel María Lacau-Mengido, Mauricio Javier Giuliodori

**Affiliations:** 1Cátedra de Higiene, Epidemiología y Salud Pública, Facultad de Ciencias Veterinarias-Universidad Nacional de La Plata (FCV-UNLP), La Plata B1900AVW, Argentina; rearteramiro@hotmail.com; 2Consejo Nacional de Investigaciones Científicas y Técnicas (CONICET), Ciudad Autónoma de Buenos Aires C1033AAJ, Argentina; dairydoc82@gmail.com (R.L.d.l.S.); bettinalacau@gmail.com (I.M.L.-M.); 3Actividad Privada, Brandsen B1890AVW, Argentina; snlorenti@gmail.com; 4Actividad Privada, Venado Tuerto S2600GOZ, Argentina; germandominguez@powervt.com.ar; 5Instituto de Investigaciones en Reproducción Animal (INIRA), Facultad de Ciencias Veterinarias-Universidad Nacional de La Plata (FCV-UNLP), La Plata B1900AVW, Argentina; 6Instituto de Biología y Medicina Experimental-CONICET, Ciudad Autónoma de Buenos Aires C1428ADN, Argentina; 7Cátedra de Fisiología, Facultad de Ciencias Veterinarias-Universidad Nacional de La Plata (FCV-UNLP), La Plata B1900AVW, Argentina

**Keywords:** body condition scoring, anestrus rate, monitoring risk factors, dairy herd

## Abstract

**Simple Summary:**

The energy status of cows during the transition period is associated with the risk of postpartum diseases, reproductive performance, and milk yield in dairy herds. The evaluation of body condition score (BCS) is a widely used tool to indirectly assess the energy balance of cows, and thresholds of frequency of cows with improper BCS have been proposed as key indicators for herd nutrition management. We evaluated the explanatory and predictive capacity of BCS indicators as risk factors for anestrus at the cow and herd levels. We found that energy balance is associated with health status, reproductive performance, and milk yield at the cow level, and that aggregated data of BCS is also associated with anestrus rate at the herd level. Despite aggregated data having a good explanatory power, their predictive capacity for anestrus rate is poor at the herd level due to the presence of other unmeasured risk factors. Therefore, we suggest that, to monitor the impact of BCS on herd disease risk, other epidemiological indicators should be used to better understand its role in productive diseases.

**Abstract:**

A retrospective longitudinal study assessing the explanatory and predictive capacity of body condition score (BCS) in dairy cows on disease risk at the individual and herd level was carried out. Data from two commercial grazing herds from the Argentinean Pampa were gathered (Herd A = 2100 and herd B = 2600 milking cows per year) for 4 years. Logistic models were used to assess the association of BCS indicators with the odds for anestrus at the cow and herd level. Population attributable fraction (AF_P_) was estimated to assess the anestrus rate due to BCS indicators. We found that anestrus risk decreased in cows calving with BCS ≥ 3 and losing ≤ 0.5 (OR: 0.07–0.41), and that anestrus rate decreased in cohorts with a high frequency of cows with proper BCS (OR: 0.22–0.45). Despite aggregated data having a good explanatory power, their predictive capacity for anestrus rate at the herd level is poor (AUC: 0.574–0.679). The AF_P_ varied along the study in both herds and tended to decrease every time the anestrous rate peaked. We conclude that threshold-based models with BCS indicators as predictors are useful to understand disease risk (e.g., anestrus), but conversely, they are useless to predict such multicausal disease events at the herd level.

## 1. Introduction

Among the multiple and interrelated metabolic challenges that dairy cows face around calving, the successful regulation of energy metabolism is mandatory to reach good productive performance throughout the whole of lactation [1]. In this sense, a substantial amount of information that explores the relationship between energy balance with disease risk, fertility, and milk yield is available [2,3]. In practice, indicators of the energy balance of cows are considered key points in most dairy herd management systems [4,5]. The strongest evidence for the negative impact of negative energy balance during the transition period on productive and reproductive aspects has mainly been evaluated at the cow level [6], but in dairy production systems most of the conditioning factors affecting energy balance that can potentially be well handled involve the herd (cow population). A herd-level analysis should provide more valuable information than a cow-level analysis for herd monitoring systems [7].

Body condition score is the most used indicator of a cow’s energy balance through external visual assessment [8]. Systematic scoring of cows at different time points during the transition period is highly recommended to identify cows that do not achieve target body conditions. The aggregation of scoring at the herd level has been suggested to detect herd energy balance problems during transition period management [4,5,9]. To design the most appropriate interventions for the transition period of dairy cows, population surveillance programs have been designed from studies that modeled the impact of energy balance indicators on productive and reproductive measures. From a methodological point of view, the scope of such modeling was exploratory when the models quantified the association between variables to understand causal relationships, whereas predictive models are required when the aim is to use some traits to predict future events [10]. As both approaches share some common methodologies, a misunderstanding of modeling could lead to the assumption that the model’s explanatory capacity and the model’s predictive capacity are the same [11,12]. To monitor energy balance at the herd level, the threshold for the frequency of cows with suboptimal energy balance indicators has been defined in multi-herd transversal studies [13,14] by accounting for the frequency of cows below a biomarker cutoff point (mostly β- hydroxybutyrate [BHB] or non-esterified fatty acids [NEFA]) measured in early lactation that is highly associated (lower *p*-value) with the frequency of a given outcome (mostly diseases or reproductive event rates). This approach gives no information about the predictive capacity of the defined herd threshold, which would be estimated by using simulation models [15] that make assumptions about the assumed prevalence of the event monitored, the sensitivity and specificity of a diagnostic test at the cow level, and the number of cows included in the analysis [16].

The objectives of the present study were 1—to assess the association of body condition score with disease risk, reproductive performance, and milk yield at the cow level; 2—to assess the predictive capacity of different proportions of cows with poor body condition indicators on the risk for anestrus at the herd level; 3—to develop a practical herd surveillance tool as to monitor the impact measures of body condition score on anestrus risk.

## 2. Materials and Methods

### 2.1. Design and Study Population

Data from two commercial grazing dairy herds from the Argentinean Pampa region were enrolled in a retrospective longitudinal study. Herd selection was performed by convenience, given that coauthors are the practitioners in charge of the reproductive management of cows in both herds. Herd A is located in Brandsen, Province of Buenos Aires (35°06′ S, 58°11′ W), with a land base of 1900 ha. It contains approximately 2100 Holstein dairy cows and has a rolling herd average milk production of around 9000 kg. Herd B is located in Carlos Casares, Province of Buenos Aires (35°37′ S, 61°22′ W), with a land base of 2000 ha. It contains approximately 2600 Holstein dairy cows, and has a rolling herd average milk production of around 11,000 kg. Healthy lactating cows were kept on a rotational system (different paddocks in the morning and afternoon) composed of mixed pastures (alfalfa, tall fescue) and winter annual grasses (ryegrass) and received concentrates (40% soybean pellets and 60% cornmeal) twice daily during milkings and partial mixed rations (corn silage, soybean pellets and cornmeal). Cows were milked twice a day (04:00 and 16:00) and milk yield was recorded during the official monthly milk test.

### 2.2. Outcome and Predictors Variables

The records of health events, reproduction, milk check and body condition score from 7.965 and 5.034 cows calving between January 2014 and December 2017 in herds A and B, respectively, were gathered. Health recorded events were metritis (defined as cows having fetid vaginal discharge < 21 DIM [17]), clinical endometritis (defined as the presence of pus in vaginal discharge > 21 DIM [18]) diagnosed by the authors (GD and NL) and clinical mastitis (defined as cows having abnormal milk secretion (e.g., clots, flakes, or watery secretion) from 1 or more quarters) [19] diagnosed by trained farm personnel in both daily milkings.

The reproductive program consisted of estrus detection assisted using tail painting (twice daily during milking), artificial insemination by farm staff and pregnancy diagnosis performed by the authors (GD and NL) using transrectal ultrasonography (at 28 to 42 d post-AI). The voluntary wait period was defined as 40 and 50 DIM in herds A and B, respectively. Anestrus was diagnosed by the authors in cows not bred within 70 DIM having no corpus luteum and a flaccid uterus by using transrectal ultrasonography and palpation. Anestrus cows were enrolled in an IATF protocol. Two dichotomic indicators were built: artificially inseminated by 80 DIM (AI80, yes vs. no) and pregnant by 100 DIM (PRE100, yes vs. no).

Body condition score was performed around calving (BCS) by farm personnel, and at the time of reproductive releases (40–60 DIM) by the authors, using a 5-point scale [8]. Both measures were used to build an indicator of body condition loss during the transition period (ΔBCS = BCS at reproductive release–BCS at calving). Data for monthly milk checks were extracted from the Official Milk Check Association. Only the data for the first monthly check were included in the analysis.

### 2.3. Statistical Analysis

Logistic regression models were fitted to assess the association of BCS indicators (BCS and ΔBCS as continuous predictors) with the odds for anestrus, metritis, mastitis, AI80 and PRE100 at the cow level, stratifying by HERD (A and B) and PARITY (1 vs. 2+). Other variables included in the models were the year of calving (2014 through 2017), and season of calving (Summer (21 December to 20 March), Autumn (21 March to 20 June), Winter (21 June to 20 Spetember) and Spring (21 Spetember to 20 December)). In the case of the COW stratum, the cow’s parity included 2, 3 and ≥4 (Equation (1)). In addition, linear regression models, stratified by HERD and PARITY, were fit to assess the association of the above-mentioned predictors with milk yield at the first monthly milk check. These linear models were also adjusted with DIM and squared DIM.
(1)$${\left({ln\left(\frac{p}{p-1}\right)}_{cow}={intercept}_{cow}+{BCS}_{cow}+{\Delta BCS}_{cow}+{Year}_{cow}+{Season}_{cow}+\varepsilon_{\text{cow}}\right)}_{\text{Herd and parity}}$$
where p is the probability of anestrus, metritis, mastitis, AI80 or PRE100 and ε is the error term at the cow level.

Logistic models were run to estimate the predictive capacity of the proportion of cows below the threshold of BCS indicators on the odds of having a high proportion of anestrus at the herd level. The entire study period was divided into 21-day cohorts based on the calving date for every herd and parity stratum. The proportion of cows with BCS < 3 or ΔBCS (loss > 0.5, Equation (2)) was estimated for every 21-day cohort, for each HERD and PARITY stratum. The quartiles from proportions for all the 21-day cohorts were calculated and were offered, one at a time, as thresholds to dichotomize each 21-day cohort (above vs. below the threshold) to predict the odds for each cohort of having a proportion of anestrus above the median. The sensibility (Se), specificity (Sp), the area under the curve (AUC) and odds ratios (OR) for each threshold were estimated with Proc Logistic of SAS 9.4, and the higher AUC was used as the selection criterium to determine the threshold at the herd level for each HERD and PARITY stratum [20].
(2)$${\left({ln\left(\frac{p}{p-1}\right)}_{21d-cohort}={intercept}_{21d-cohort}+{BCS-\Delta BCS}_{21d-cohort}+\varepsilon_{21d-cohort}\right)}_{\text{Herd and parity}}$$

Logistic model where p is the probability that a 21-day cohort have an anestrus proportion above the median, BCS − ΔBCS is the proportion of cows with BCS < 3 or ΔBCS > 0.5 above or below the threshold and ε is the error term at the 21-day cohort level.

Finally, the population attributable fraction (AF_P_) of anestrus rate due to body condition indicators was estimated for each 21-day cohort at every HERD and PARITY stratum. The AF_P_ estimates how much the proportion of anestrus would be reduced in a population if none of the cows would have been exposed to the risk factor (BCS < 3 or ΔBCS loss > 0.5), assuming a causal relationship [21]. The AF_P_ was estimated with the following formula:$${AF}_{P}=pd\left(\frac{aRR-1}{aRR}\right)$$
where pd is the proportion of anestrus that had BCS < 3 or ΔBCS loss > 0.5 and aRR is the adjusted Risk Ratio. The aRR was estimated by adjusting stratified (by 21-day cohorts) logistic models, for each HERD and PARITY stratum, with BCS indicators (BCS < 3 or ΔBCS loss > 0.5; yes/no) as the main categorical predictors adjusted by calving year and calving season. In the case of the cow stratum, the parities included were 2, 3, and 4+. Finally, a time series was built for anestrus risk and AF_P_ including each 21-day cohort at each HERD and PARITY stratum for all the study periods.

## 3. Results

Descriptive data about anestrus, metritis, mastitis, AI80, PRE100 and milk for both herds and parity groups are shown in Table 1. Descriptive data about BCS and ΔBCS by HERD and PARITY stratum are shown in Figure 1. Cows and heifers from herd A had a higher BCS at calving and showed less variability regarding ΔBCS than those from herd B. 

The association between BCS and ΔBCS with anestrus, metritis, mastitis, AI80 and PRE100 at the cow level is shown in Table 2. Logistic models showed that cows and heifers with higher BCS at calving and smaller ΔBCS had a lower risk for anestrus and metritis and a higher risk for being AI80 and PRE100. In addition, the effect size (OR) was similar across herd and parity strata. Regarding the association between BCS indicators (BCS and ΔBCS) with milk yield at the first monthly milk check, we found that cows and heifers calving with better BCS produced more milk than herd mates with poorer BCS in herd A, but that association was not observed in herd B. Additionally, cows from both herds and heifers from herd B having a higher ΔBCS (BCS loss) produced more milk than herdmates that were not losing BCS. Finally, the BCS indicator was not associated with clinical mastitis in any herd or parity category.

The logistic model estimating the predictive capacity of the proportion of cows below the threshold of BCS indicators for every 21-day calving cohort on the odds of a cohort having a proportion of anestrus over the median at the herd level is shown in Table 3. This model (BCS < 3 or ΔBCS [loss > 0.5]) showed that the best threshold to predict a cohort having a risk of anestrus over the expected median in cows was the bottom quartile (Q_25_ = 20%) in herd A and the top quartile (Q_75_ = 78%) in herd B (Table 3). In the case of heifers, the best threshold to predict having a risk of anestrus over the expected median was the middle quartile (Q_50_ = 6% and 74% for herds A and B, respectively [Table 3]). The results of the ROC curve analysis are also shown in Table 3. The AUC varied from 0.574 to 0.679 depending on the herd and parity category.

Figure 2 shows a time series for anestrus risk and AF_P_ including each 21-day cohort at each HERD and PARITY stratum for all the study periods. We found that the anestrus rate was higher in herd B than in herd A for both parity groups (heifers’ median = 9.4% vs. 15.7% for herds A and B, respectively; cows’ median = 11.6% vs. 20.7% for herds A and B, respectively), and also that the variability in anestrus rate was higher in heifers than in cows (interquartile range were 12.9% and 18.9% for heifers in herds A and B, respectively, and they were 7.0% and 10.9% for cows in herds A and B, respectively). We also found a higher FA_p_ for both cows and heifers in herd A than in herd B, and that cows had a higher FA_p_ than heifers in both herds. Finally, we found that the FA_p_ decreased as the anestrus rate increased.

## 4. Discussion

Our findings that the cows and heifers calving with low BCS or losing much BCS postpartum are those at the highest risk for disease (e.g., anestrus and metritis) and with the poorest fertility at the cow level agree with previous reports [2,3,6]. The explanation for the association between BCS indicators and disease risk and fertility is that transition dairy cows are not able to consume enough energy to fulfill their increased requirements, which makes them susceptible to experiencing a delayed uterine involution and/or a delayed resumption of ovarian activity postpartum [22,23,24]. In turn, delayed uterine involution and/or delayed cyclicity have been associated with a higher risk for uterine diseases such as metritis and endometritis, and, also, with anestrus [25,26,27]. It is worth mentioning that, despite the large variability we observed in BCS at calving and ΔBCS postpartum, the direction and the magnitude of the associations (e.g., OR) of BCS indicators with disease risk and fertility were similar for both herds and parity category (Figure 1 and Table 2). Therefore, our results are in line with Friggens et al. [28], who proposed that, independently of the BCS at calving, dairy cows will return to a target BCS during the postpartum period. That is, fatter cows at calving would lose more BCS than their thinner herd mates. In this sense, the rate of lipid mobilization and the size of the lipid reserve are considered the key determiners of the resumption of postpartum cyclicity in dairy cows [28]. Our results (Figure 1) clearly showed that cows and heifers in herd A have higher BCS at calving and, also, that they lose more BCS postpartum than cows and heifers in herd B. 

Regarding the association at the herd level, we also found a strong association between the 21-day cohorts with a high frequency of cows having poor BCS indicators and the anestrus rate (Table 3). These results are in line with previous reports assessing other indicators of the energy balance, like NEFA and BHBA [13,14]. Chapinal et al. [13] found a strong association (OR: around 2) between herd-level thresholds of 5 to 50% of the sampled cows with increased NEFA (≥0.50 and ≥1.0 mEq/L pre- and postpartum, respectively) or BHBA (≥0.80 mEq/L) with a higher risk of disease (e.g., displaced abomasum) and poorer productive and reproductive performances. Ospina et al. [14] also detected a good association between herd-level thresholds of 15 to 20% of the sampled cows with high prepartum NEFA (≥0.27 mEq/L), high postpartum NEFA (≥0.60 mEq/L) and BHBA (≥1.00 mEq/L) with increased risk of disease (e.g., displaced abomasum and clinical ketosis) and reduced milk production and pregnancy rates. 

Surprisingly, these same logistic models that have shown such a strong association between BCS indicators and the anestrous rate at the herd level (OR from 0.220 to 0.452) showed a very poor predictive capacity given that the AUC varied from 0.574 to 0.673 (Table 3). Therefore, it can be pointed out that BCS frequency distribution, as a cohort threshold, seems inadequate to predict an increase in anestrus rate. That is, the monitoring of BCS is worthy to understand the dynamics of energy balance at the herd level, but it would not be useful to predict the future performance of the herd (e.g., anestrous rate). A possible explanation for this lack of predictive capacity could be the multi-causal nature of anestrus. It is worth pointing out that an epidemiological impact measure like the AF_p_, based on association measures (e.g., relative risk), could be useful to explain the observed results. In this sense, we could state that 50–60% of anestrus events in herd A could be avoided if none of the cows had poor BCS (assuming a causal relationship between BCS and anestrus, [21]). Additionally, when the anestrus rate increased to a peak of 0.35 for 21-day calving cohort (number 15) in cows of herd A (Figure 2A), the AF_p_ had a low value of 0.3, suggesting that other factors could explain this increase in punctual anestrus rate. The same can be seen for 21-day calving cohort (number 15) in heifers of herd A (Figure 2B), which had an anestrous rate peak at 0.27 accompanied by a low AF_p_ value of 0.3. In the case of herd B, most of the time the anestrus rate was explained by the poor BCS in cows and heifers because the AF_p_ value was high (around 1.0, Figure 2), but every time the anestrus rate increased, the AF_p_ tended to decrease.

As already mentioned, there is a general agreement that the dynamics of energy balance along the lactation cycle has a big impact on the health status and the reproductive and productive performance of dairy cows. Therefore, some indirect indicators have been proposed to monitor energy balance to make decisions regarding the nutritional management of the herd. Previous transversal observational studies have measured different indicators (NEFA or BHBA) during the transition period from multiple herds to define the herd’s threshold for the frequency of cows under a given cutoff point [13,14]. These previous works estimated the proportion of cows with poor energy balance that have the strongest association with the incidence of a given outcome (mostly a disease or pregnancy rate) based on the *p*-value of that association. Therefore, the herd threshold was selected depending on the statistical significance of the association (on the probability of type I error), without considering the predictive capacity of the herd threshold [13,14]. From a methodological point of view, the approach applied in the present work has two main differences: first, the herd threshold was defined based on their predictive capacity (not on their explanatory power), and second, it involved a longitudinal study where the counterfactual groups were cohorts belonging to the same herds. This aspect would reduce the likelihood of herd-level confounders in the model’s estimates [29]. The latter two aspects, assessing the model’s predictive capacity, and using specific intra-herd data to build the model, would strengthen the link between scientific knowledge acquisition and its practical application. According to Shmugli [11], it is key to differentiate between the explanatory power and predictive power of statistical models because they are commonly confounded. Therefore, it is very important to consider the explanatory and predictive powers of statistical models used to make decisions in practice [11]. According to our findings, the cut-off of BCS indicators at the herd level has a poor predictive capacity for anestrus rate. Despite these models having a strong explanatory capacity at the cow level, the use of fixed thresholds as a criterion for monitoring energy balance at the herd level is debatable. The idea behind the use of fixed cut-off values to monitor the herd’s energy balance status is very attractive, given its simplicity and potential practical application. However, as the cut-off determination has mainly been based on causal models (poor energy balance leading to high disease risk and low productive performance) without considering their predictive capacity, the likelihood for them to be inaccurate tools increases. Therefore, the use of epidemiological indicators, like AF_P_, could provide decision-makers with a practical interpretation of the impact that poor energy balance indicators have on the risk of disease.

Beyond methodological aspects, the predictive capacity of these models could be improved by combining information from precision dairy tools (like sensor-obtained data) with the described energy balance indicators (BCS, NEFA, and BHBA). Additionally, it could be improved by changing the time frame when these indicators are measured (e.g., prepartum and postpartum). In this sense, Wisniesky et al. [30,31,32] performed prospective cohort studies in five dairy herds to assess the use of nutritional (NEFA, BHBA, and Ca), oxidative (ROS and TAC), and inflammatory (haptoglobin, SAA, and WBC count) biomarkers at the dry-off and other covariates as predictors of transition diseases at the cow and cohort levels. These researchers found a high predictive capacity at the cow level, and, also, an acceptable predictive capacity at the cohort level. This suggests that, given the multi-causal nature of health status and reproductive and productive performance, building models with a higher number of predictors, measured in a different time (e.g., prepartum), and, with proper individual cow data, aggregation could improve the predictive performance of these herd monitoring strategies. 

This retrospective cohort study was carried out in two commercial dairy herds, selected by convenience, that have similar health and reproductive management but different nutritional management, which explains the observed differences in the frequency distribution of energy balance indicators and anestrus rates. One of the limitations is that, with the available data, it is not possible to assess the agreement coefficient between BCS evaluators. Therefore, there is a possibility for misclassification bias of this predictor variable. We assume that, if this misclassification of exposure (BCS) occurs, it is independent of outcome (anestrus). Therefore, this non-differential error would bias the association toward to null. As this study was designed to achieve robust internal validity by defining thresholds using historical data from each herd, one limitation is that the scope of the inference reaches only these two herds and, therefore, these results should not be extrapolated to other dairy cow populations. 

One of the implications of this study is that, to predict multifactorial events like diseases and productive performance, models should include a higher number of predictors (e.g., biomarkers of nutritional, oxidative, and inflammatory status) to improve the predictive power of these herd monitoring strategies. Another implication is that herd monitoring predictor strategies should use epidemiological indicators (like AF_P_) temporally associated with disease risk that are estimated from data gathered in the same herds where the models are applied. Therefore, the approach used in the present study offers a valid alternative for the herd monitoring of energy balance to the fixed threshold approach previously proposed in the bibliography. 

## 5. Conclusions

We found that BCS is associated with health status, reproductive performance, and milk yield at the cow level, and that aggregated data of BCS at the herd level are associated with anestrus rate, but its predictive capacity for anestrus rate at the herd level is poor. We conclude that threshold-based models with BCS indicators as predictors are useful for understanding disease risk (e.g., anestrus), but conversely, they are useless at predicting such multicausal disease events at the herd level. 

## Figures and Tables

**Figure 1 animals-13-03114-f001:**
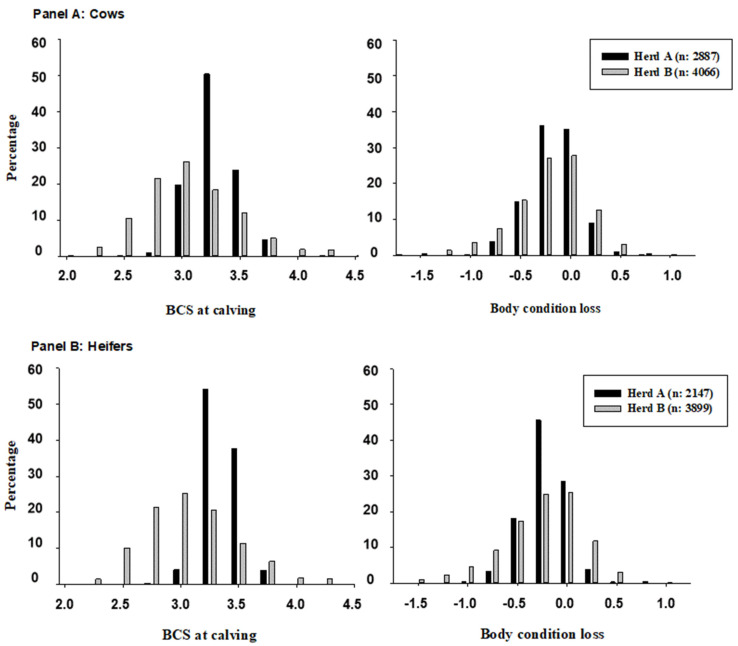
Frequency of body condition score at calving and body condition loss during the transition period in cows and heifers belonging to the herd A and B.

**Figure 2 animals-13-03114-f002:**
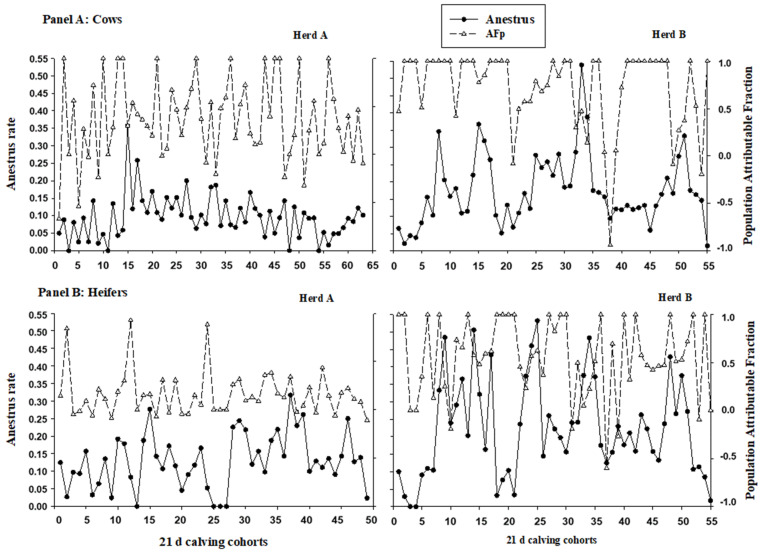
The anestrus rate and population attributable fraction (AF_P_) by 21-day cohorts for each herd and parity group from January 2014 to October 2017.

**Table 1 animals-13-03114-t001:** Descriptive data about anestrus, metritis, mastitis, cows inseminated by 80 DIM, cows pregnant by 100 DIM and first monthly milk check for both herds and parity groups from January 2014 to October 2017.

Herd	Parity (n)	Anestrus ^1^ (%)	Metritis ^2^ (%)	Mastitis ^3^ (%)	PRE100 ^4^ (%)	AI80 ^5^ (%)	Milk ^6^ (Mean)
A	Heifers (2.147)	14.0	22.8	7.1	31.9	64.9	24.8
	Cows (2.887)	10.4	26.1	14.0	29.0	70.0	29.9
B	Heifers (3.899)	23.5	27.0	30.0	31.7	61.2	24.7
	Cows (4.066)	16.5	12.3	28.1	20.5	58.9	33.6

^1^ Anestrus: defined as the absence of corpus luteum together with a flaccid uterus by using transrectal ultrasonography and palpation. ^2^ Metritis: defined as cows having fetid vaginal discharge <21 DIM. ^3^ Mastitis: defined as cows having abnormal milk secretion (e.g., clots, flakes, or watery secretion) from 1 or more quarters. ^4^ PRE100: defined as pregnant cows by 100 DIM (yes vs. no). ^5^ AI80: defined as inseminated cows by 80 DIM (yes vs. no). ^6^ Milk: milk yield (kg/d) at first monthly milk check (30–40 DIM).

**Table 2 animals-13-03114-t002:** Multivariable logistic and linear models assessing the associations of body condition score (BCS) at calving and the body condition change of up to 30–40 DIM (ΔBCS) with the odds for anestrus, metritis, mastitis, being inseminated by 80 DIM (AI80), being pregnant by 100 DIM (PRE100) and with milk yield at first monthly milk check (30–40 DIM).

					OR (95%CI) ^1^			
Herd	Parity	Predictor	Anestrus ^2^	Metritis ^3^	Mastitis ^4^	PRE100 ^5^	AI80 ^6^	Milk ^7^
A	Heifers (*n* = 2.147)	BCS ^8^	0.07 (0.04–0.12)	0.31 (0.21–0.48)	1.12 (0.60–2.07)	2.35 (1.63–3.41)	4.48 (3.11–6.44)	1.33 (0.82)
		ΔBCS ^9^	0.36 (0.30–0.42)	0.60 (0.52–0.70)	0.75 (0.60–0.93)	1.49 (1.31–1.70)	1.89 (1.66–2.16)	0.18 (0.58)
	Cows (*n* = 2.887)	BCS	0.05 (0.04–0.08)	0.40 (0.31–0.53)	0.96 (0.70–1.30)	1.67 (1.30–2.14)	3.09 (2.37–4.02)	2.59 (0.80) *
		ΔBCS	0.33 (0.28–0.39)	0.69 (0.62–0.76)	0.94 (0.83–1.08)	1.43 (1.28–1.59)	1.72 (1.54–1.93)	−2.53 (0.68) *
B	Heifers (*n* = 3.899)	BCS	0.13 (0.11–0.17)	0.83 (0.70–0.99)	0.90 (0.79–1.03)	2.02 (1.76–2.31)	3.08 (2.65–3.58)	−0.68 (0.41)
		ΔBCS	0.41 (0.38–0.46)	0.83 (0.76–0.91)	1.02 (0.95–1.08)	1.32 (1.23–1.41)	1.62 (1.51–1.74)	−2.24 (0.41) *
	Cows (*n* = 4.066)	BCS	0.12 (0.10–0.15)	0.67 (0.56–0.79)	0.97 (0.87–1.08)	1.42 (1.25–1.62)	2.20 (1.94–2.48)	−0.01 (0.46)
		ΔBCS	0.37 (0.34–0.42)	0.70 (0.45–1.10)	1.00 (0.94–1.06)	1.13 (1.06–1.21)	1.42 (1.34–1.52)	−1.46 (0.49) *

Logistic models were controlled by year of calving (2014 through 2017), the season of calving (Summer (21 December to 20 March), Autumn (21 March to 20 June), Winter (21 June to 20 Spetember) and Spring (21 Spetember to 20 December)), and also by parity (2, 3 and ≥4) in the cow models. Data on the year of calving (*p* < 0.001), the season of calving (*p* < 0.001), and parity (*p* < 0.001) are not shown in Table 2. ^1^ OR (95% CI): Odds ratio (and confidence intervals) were estimated with the Proc Glimmix of SAS (odds estimated per unit of increase over the mean). ^2^ Anestrus: defined as the absence of corpus luteum together with a flaccid uterus by using transrectal ultrasonography and palpation. ^3^ Metritis: defined as cows having fetid vaginal discharge <21 DIM. ^4^ Mastitis: defined as cows having abnormal milk secretion (e.g., clots, flakes, or watery secretion) from 1 or more quarters. ^5^ PRE100: defined as pregnant cows by 100 DIM (yes vs. no). ^6^ AI80: defined as inseminated cows by 80 DIM (yes vs. no). ^7^ Milk estimates (and Standard Error) were obtained from the lineal model. ^8^ BCS = Body condition score (5-point scale) at calving as a continuous predictor. ^9^ ΔBCS = BCS at reproductive release–BCS at calving as a continuous predictor. * *p* < 0.001.

**Table 3 animals-13-03114-t003:** Estimation of sensibility, specificity, the area under the curve, and odds ratios of herd threshold for frequency of low body condition at calving (BCS < 3) or high body condition loss up to 30–40 DIM (ΔBCS -> 0.5), by 21-day periods, as a predictor of anestrus cow rate over the median of herd and category.

Herd	Parity	Threshold ^1^	OR (95%CI) ^2^	Se ^3^	Sp ^4^	AUC ^5^
A	Heifers ^6^	Q_50_ (6%)	0.22 (0.07–0.76)	69.2%	66.6%	0.679
A	Cows ^7^	Q_25_ (20%)	0.45 (0.14–1.47)	81.5%	33.3%	0.574
B	Heifers ^8^	Q_50_ (74%)	0.25 (0.06–0.73)	66%	68%	0.673
B	Cows ^9^	Q_75_ (78%)	0.29 (0.08–1.11)	37%	85%	0.611

^1^ Threshold: Herd quartile (25 vs. 50 vs. 75) representing the threshold and percentage of individuals with BCS < 3 or ΔBCS -> 0.5 from 21-day cohorts (to be included in the study, number of heifers or cows by 21-day cohort should be ≥20). ^2^ OR (95%CI): Odds ratio and confidence intervals. ^3^ Se: sensitivity. ^4^ Sp: specificity. ^5^ AUC: area under the curve. ^6^ Median of heifers by cohort in herd A = 35 (inter quartile range = 21–52; cohort n = 47). ^7^ Median of cows by cohort in herd A = 47 (interquartile range= 36–61; cohort n = 62). ^8^ Median of heifers by cohort in herd B = 67 (interquartile range = 57–83; cohort n = 55). ^9^ Median of cows by cohort in herd B = 76 (interquartile range= 62–94; cohort n = 54).

## Data Availability

The aggregated data presented in this study are openly available in Zenodo repository. Data at doi:10.5281/zenodo.8082738, accessed on 26 June 2023.

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
