# Peer review of "Monitoring of Body Condition in Dairy Cows to Assess Disease Risk at the Individual and Herd Level"

_animals, 2023, doi:10.3390/ani13193114_

Round 1

Reviewer 1 Report

To the authors:

This study evaluated the association of body condition score losses pre and post calving with metritis and also anestrus in dairy cattle by parity. The study is interesting, adequately powered, correctly statistically assessed, and of interest to the journal. The authors need to be a bit more transparent about the limitations of the study. Firstly, that this applied to dairy cattle in a region where heat stress is a common issue that may be causing more anestrus than is representative of dairy cattle in other parts of the world, and second, that there was no formal measures of agreement performed between the co-workers and co-authors for measuring the primary outcome: BCS. Body condition score can vary greatly among researchers due to bias, it is difficult to train agreement across observers. Therefore, there is a possibility that some unintended effects of parity biased the BCS pre and post calving. The authors should acknowledge that this is a limitation to the study.

Author Response

Response to Reviewer 1 Comments

Point 1: This study evaluated the association of body condition score losses pre and post calving with metritis and also anestrus in dairy cattle by parity. The study is interesting, adequately powered, correctly statistically assessed, and of interest to the journal.

Response 1: Thanks for your positive comments!

Point 2: The authors need to be a bit more transparent about the limitations of the study. Firstly, that this applied to dairy cattle in a region where heat stress is a common issue that may be causing more anestrus than is representative of dairy cattle in other parts of the world, and second, that there was no formal measures of agreement performed between the co-workers and co-authors for measuring the primary outcome: BCS. Body condition score can vary greatly among researchers due to bias, it is difficult to train agreement across observers. Therefore, there is a possibility that some unintended effects of parity biased the BCS pre and post calving. The authors should acknowledge that this is a limitation to the study.

Response 2: The Reviewer 1 made good points about the study limitations. Regarding heat stress, we don’t have any direct measure of it. To avoid a reduction in internal validity we included season of parity as a covariate in the model. We agree with the Reviewer 1 that heat stress could be another aspect reducing the extrapolation of results to other populations.

Regarding measurement of BCS, we do not have the agreement coefficient between co-workers and co-authors evaluating BCS. Therefore, if that is the case, there is a possibility for misclassification of the predictor variable that would lead to a lower internal validity. The impact of this likely bias is hard to control and quantify, but we know that the estimated association would only be biased if a systematic misclassification of BCS were dependent on anestrus risk. A priori we cannot find a reason why the misclassification of exposure (BCS) and outcome (anestrus) are not independent. Therefore, we do not expect our estimations are hardly biased and in case of occurrence the measure of association would bias toward the null. This point was added to the revised manuscript as a study limitation.

Reviewer 2 Report

The research enhances our understanding of the effects of postpartum body condition in dairy cows. The article's topic is fascinating, and I personally read the manuscript with great interest. The paper aligns well with the scope of the journal. However, in its current form, I believe it has several shortcomings:

-          The title is too long. I suggest selecting a shorter version.

-          I suggest rewriting the abstract to make it clearer and include more results and discussions.

-          The Material and Methods section needs to be expanded to provide more details and enhance the reproducibility of the research. Additionally, I suggest clearly reporting the equation models used for the analysis to facilitate the reader's understanding of the statistical methods adopted.

-          I suggest rewriting the Results section to improve readability and include more data. It would be beneficial to insert some descriptive statistics of the animals involved in the research within the Results section.

-          It would be beneficial to include practical implications and study limitations in the Discussion section

Specific comments:

Lines 62-64: Please insert a citation. For example, I suggest reading and citing the paper with the doi 10.1080/1828051X.2022.2032850.

Lines 68-71: Please insert a citation. For example, I suggest reading and citing the paper with the doi 10.3389/fvets.2023.1141286.

Line 94: Please add information about the feeding management of the herds involved in the study.

Line 101: How many cows were involved in the study?

Author Response

Response to Reviewer 2 Comments

Point 1: The research enhances our understanding of the effects of postpartum body condition in dairy cows. The article's topic is fascinating, and I personally read the manuscript with great interest. The paper aligns well with the scope of the journal.

Response 1: Thanks for your positive comments!

However, in its current form, I believe it has several shortcomings:

Point 2: The title is too long. I suggest selecting a shorter version.

Response 2: We shortened the title as follows: Monitoring of body condition in dairy cows to assess disease risk at the individual and herd level.

Point 3: I suggest rewriting the abstract to make it clearer and include more results and discussions.

Response 3: We revised the abstract trying to follow the Reviewer 2’s comments. Anyway, we could not expand so much due to 200 words limitation set in the instruction for authors.

Point 4: The Material and Methods section needs to be expanded to provide more details and enhance the reproducibility of the research. Additionally, I suggest clearly reporting the equation models used for the analysis to facilitate the reader's understanding of the statistical methods adopted.

Response 4: The Material and Methods section was expanded and statistical model equations were included as suggested.

Point 5: I suggest rewriting the Results section to improve readability and include more data. It would be beneficial to insert some descriptive statistics of the animals involved in the research within the Results section.

Response 5: Descriptive data about anestrus, metritis, mastitis, Inseminated cows by 80 DIM, pregnant cows by 100 DIM, and first monthly milk check for both herds and parity groups were included as a table at the beginning of the Result section.

Point 6: It would be beneficial to include practical implications and study limitations in the Discussion section.

Response 6: As suggested by Reviewer 2, we added practical implications and study limitations to the discussion section.

Specific comments:

Point 7: Lines 62-64: Please insert a citation. For example, I suggest reading and citing the paper with the doi 10.1080/1828051X.2022.2032850.

Response 7: Citation added as suggested.

Point 8: Lines 68-71: Please insert a citation. For example, I suggest reading and citing the paper with the doi 10.3389/fvets.2023.1141286.

Response 8: Citation added as suggested.

Point 9: Line 94: Please add information about the feeding management of the herds involved in the study.

Response 9: Feeding management was added to the revised manuscript as suggested by Reviewer 2.

Point 10: Line 101: How many cows were involved in the study?

Response 10: That information was added to the revised manuscript.

Reviewer 3 Report

This manuscript describes an experiment in which retrospective data was used to examine the relationship between various factors and likelihood of anoestrus at both cow and herd level.

The cow level aspects of this paper seem mostly well written and justified.  It is a common problem in bovine heart health that whilst relationships are often well understood, there is too much variation for the relationships to be predictive and the authors have done a good job of explaining that.

I'm very unsure about the herd level conclusions since there are only two herds and there is no evidence that the results same here might apply to other herds.  I hold the view that whilst it is worth pointing out the herds were different, and they were different in a way we might expect, I don't believe that the analysis about predictive value for these two herds is useful or appropriate.

I also had some concerns about the conclusions.  The conclusions read like the authors are the first people to have associated transition period energy balance with health status reproductive performance and milk yield at the cow level.  I think the conclusions should say that our results were consistent with previous work that shows such relationships, and that their attempts using large numbers of cows to make a predictive model we're not successful due to the wide variation that exists.  The comment about the use of impact measures like AFP providing risk factors at the herd level is probably true but I do not think that is a conclusion of the data which involved only two Herds.

Author Response

Response to Reviewer 3 Comments

Point 1: This manuscript describes an experiment in which retrospective data was used to examine the relationship between various factors and likelihood of anoestrus at both cow and herd level.

The cow level aspects of this paper seem mostly well written and justified.  It is a common problem in bovine heart health that whilst relationships are often well understood, there is too much variation for the relationships to be predictive and the authors have done a good job of explaining that.

Response 1: Thanks for your positive comments!

Point 2: I'm very unsure about the herd level conclusions since there are only two herds and there is no evidence that the results same here might apply to other herds. I hold the view that whilst it is worth pointing out the herds were different, and they were different in a way we might expect, I don't believe that the analysis about predictive value for these two herds is useful or appropriate.

Response 2: The conclusions were rewritten. We stated clearly that extrapolation to other dairy cow population should not be done given that the study was performed on data gathered from two herds selected by convenience. That aspect was included as a study limitation in the discussion section.

Point 3: I also had some concerns about the conclusions. The conclusions read like the authors are the first people to have associated transition period energy balance with health status reproductive performance and milk yield at the cow level. I think the conclusions should say that our results were consistent with previous work that shows such relationships, and that their attempts using large numbers of cows to make a predictive model we're not successful due to the wide variation that exists. The comment about the use of impact measures like AFP providing risk factors at the herd level is probably true but I do not think that is a conclusion of the data which involved only two Herds.

Response 3: As already said in response to point 2, the conclusions were rewritten. It was not our intention to pretend that we were the first saying transition period energy balance is associated with health status, productive performance, and reproductive performance. We agree that the large variability observed reduces the predictive capacity of these models using only one indicator of BCS as predictor. Therefore, this lack of predictive capacity is one of the main drawbacks to define the thresholds for monitoring BCS. That is, how do we define the cut-off and what for?

Round 2

Reviewer 2 Report

The authors have done a good job, and now the paper has been greatly improved. I just bring the authors' attention to Line 188, where there is the beginning of a sentence without the ending.

Author Response

Thanks for your positive comments! We revised the sentence. It was our mistake. We deleted that incomplete sentence.